# Influence of Surgical Pleth Index-Guided Versus Conventional Analgesia on Opioid Consumption During Gastric Sleeve Surgery: A Pilot Study

**DOI:** 10.3390/life15101570

**Published:** 2025-10-08

**Authors:** Crina-Elena Leahu, Cristina Petrisor, Simona Cocu, Alexandra Maria Boldis, George Calin Dindelegan

**Affiliations:** 1Department of Surgery, Discipline of Surgery, “Iuliu Hatieganu” University of Medicine and Pharmacy, 8 Victor Babes Street, 400012 Cluj-Napoca, Romania; 2First Surgical Clinic, Clinical Emergency County Hospital, 3-5 Clinicilor Street, 400347 Cluj-Napoca, Romania; 3Department of Surgery, Discipline of Anesthesia and Intensive Care, “Iuliu Hatieganu” University of Medicine and Pharmacy, 8 Victor Babes Street, 400012 Cluj-Napoca, Romania

**Keywords:** nociception monitoring, SPI (surgical plethysmography index), gastric sleeve surgery, opioid consumption

## Abstract

Recent advances in intraoperative nociception monitoring, such as the Surgical pleth index (SPI, GE Healthcare, Helsinki, Finland), may help optimize opioid use. Obese patients are particularly susceptible to opioid-related side effects, making this approach of interest in bariatric surgery. In this randomized pilot study, we investigated whether SPI-guided fentanyl administration would influence intraoperative opioid use and postoperative pain. We enrolled 49 patients undergoing laparoscopic gastric sleeve surgery under sevoflurane-based general anesthesia with multimodal perioperative analgesia, randomized to conventional fentanyl dosing at the anesthetist’s discretion (*n* = 25) or SPI-guided dosing (*n* = 24). The primary endpoint was intraoperative fentanyl consumption. Secondary outcomes included time to extubation, hemodynamic events, pain scores in the first 90 min postoperatively and rescue analgesia. Fentanyl use did not differ significantly between groups (SPI: 400 ± 101 mcg vs. control: 450 ± 56 mcg, *p* = 0.100). Extubation was faster with SPI guidance (8.1 ± 1.6 vs. 9.6 ± 1.3 min, *p* < 0.001). Hemodynamic events and rescue analgesia were less frequent in the SPI group, though not statistically significant. Pain scores were comparable, and no opioid-related adverse effects occurred. In our study, SPI-guided opioid administration did not reduce overall intraoperative fentanyl requirements compared with conventional practice but was associated with a modestly shorter time to extubation.

## 1. Introduction

Intraoperative monitoring has evolved in recent years, from the minimal, obligatory parameters of ECG (electrocardiography), SpO_2_, NIBP (non-invasive blood pressure) and capnography [1,2], to also monitoring of depth of anesthesia and nociception [3]. Whereas pain is the subjective, unpleasant sensory and emotional experience associated with actual or potential tissue damage [4], nociception is defined as the physiological encoding and processing of nociceptive stimuli [5], making it more difficult to quantify, especially in unconscious patients undergoing general anesthesia. Several methods have been proposed, some relying on variations in sympathetic/parasympathetic tone (Surgical pleth index (SPI), pupillometry, Analgesia nociception index monitor (ANI, Mdoloris Medical Systems, Loos, France), Nociception level index, Skin conductance), others on EEG (electroencephalography) changes (qNOX-qCON 2000 Monitor, Quantium Medicaln, Mataro, Spain, Entropy monitoring: response entropy) [5,6]. A recent meta-analysis of randomized controlled trials [7] comparing different nociception monitors with standard care or versus other nociceptor devices during general anesthesia has failed to prove a reduction in opioid consumption associated with their use, except for pupillometry. There were also no differences in postoperative pain scores, rescue analgesia and postoperative nausea and vomiting (PONV). However, nociception monitoring remains a focus of research because it promotes personalized anesthesia and opioid-sparing goals.

SPI is derived from the pulse oximeter wave, using the heartbeat interval and the photoplethysmographic waveform amplitude to calculate an index that shows the balance between nociceptor activation and analgesia during general anesthesia [8]. It is easy to use and interpret as it only requires that the pulse oximeter be attached to the finger. Values range from 0 to 100, with higher values denoting a more intense nociceptive stimulus [9,10]. A threshold value of 50 sustained for more than 3–5 min has previously been used as an indication to administer additional opioid doses during general anesthesia [11,12]. So far, SPI has been validated as an adjunct to guiding intraoperative opioid administration [13,14] and predicting postoperative pain [2,15,16]. It has also shown good correlation with stress hormones response during surgery [17]. However, because it relies on autonomic responses, there are also several factors that can affect its reliability and interpretation during anesthesia [5,18]. These confounders can be patient-related (patients with neuropathies and autonomic dysfunction may show altered responses), anesthesia-related (deeper anesthesia can dampen autonomic reactivity, administration of atropine, beta-mimetics or beta-blockers influences the relationship between the sympathetic and parasympathetic systems), physiological factors (hypovolemia, vasoconstriction, hypothermia may influence the plethysmography pulse wave) or external factors (lithotomy position may decrease SPI significantly [19]).

Bariatric surgery plays a key role in the treatment of obesity and obesity-related pathology, having recently extended its indications [20]. Anesthesia for the obese patient has its challenges concerning both respiratory and cardiovascular changes, as well as the pharmacology of drugs. Obese patients have reduced functional residual capacity, hypoventilation and sleep-disorder breathing that make them more susceptible to the blunting effect of opioids on the respiratory drive, greatly increasing the risk of hypoxemia, apnea and respiratory arrest in the perioperative period. Pharmacokinetic challenges include altered volume of distribution for lipophilic drugs, including opioids, potentially leading to drug accumulation and prolonged sedation. In current practice lean body weight, reflecting the fat-free mass, is usually used to calculate drug doses for most drug classes [21,22,23,24]. A multimodal approach to guiding analgesia perioperatively has been proposed, with recommendations to use opioid-sparing approaches to improve postoperative recovery and reduce the incidence of complications in these patients [25].

There is no current data on nociception monitoring using SPI strictly in obese patients. The aim of this study is to determine whether there is any difference between intraoperative opioid consumption when guiding opioid administration by SPI versus conventional analgesia provided by the anesthetist.

## 2. Materials and Methods

This randomized control trial received approval from the ethics committees of the Cluj-Napoca County Hospital (No. 57290/20 December 2022) and the University of Medicine and Pharmacy “Iuliu Hatieganu” Cluj-Napoca (No. AVZ59/12 April 2023) and was prospectively registered on clinicaltrials.gov (ID NCT05884229).

### 2.1. Study Population

We enrolled 49 patients, between May 2023–June 2024, undergoing laparoscopic sleeve gastrectomy in our department. All patients included in our study underwent bariatric surgery based on established clinical indications (BMI ≥ 40 kg/m^2^ alone or with BMI ≥ 35 kg/m^2^ and additional comorbidities) and multidisciplinary evaluation at our institution. Recruitment was performed after gaining written informed consent from the patients.

### 2.2. Study Design

The enrolment was prospectively conducted and the flowchart is summarized in Figure 1. Inclusion criteria consisted of age 18–65 years, no gender restrictions, ASA score II–III, patient’s informed consent. Exclusion criteria were medication that affects the autonomous nervous system, peripheral neuropathy, abnormal renal or liver function, history of opioid abuse, atropine or vasoactive medication administration during surgery. For sample size estimation we preliminarily enrolled 5 patients in each group. We used the data for calculating the sample size for detecting a difference of 50 mcg of fentanyl with a significance level of 5% and statistical power of 80%.

### 2.3. Anesthesia

Patients were randomly allocated into one of the two groups using https://www.random.org: one group with conventional opioid administration, independent of SPI, as considered by the anesthetist (the control group), and the second group with SPI-guided opioid dosing (the interventional group), as detailed further.

We recorded baseline vital signs—heart rate (HR) and mean arterial pressure (MAP)—both on the ward and at admission to the operating room. The patients received standard monitoring: 5-derivation ECG, blood pressure (either non-invasively (NIBP) or invasively in those patients in which NIPB was not reliable owing to too small cuffs), pulsoximetry (Carescape TM monitor ^®^B650, GE Healthcare, Helsinki, Finland), end-tidal CO_2_, train-of-four (TOF), bispectral index (BIS). General anesthesia was induced using 1% Lidocaine 1 mcg/LBW, fentanyl 2 mcg/LBW (lean body weight), propofol 2 mg/LBW, rocuronium 1 mg/kg LBW and patients were intubated in a rapid sequence induction. Anesthesia was maintained using sevoflurane targeting BIS 40-60. Mechanical ventilation was performed protectively, with inspiratory volumes of 6 mL/kg predicted body weight (PBW), FiO_2_ 60% and respiratory rate adjusted for maintaining PaCO_2_ 35–40 mmHg. Intraoperative multimodal analgesia consisted of paracetamol 1 g before incision, low dose ketamine 0.3 mg/kg LBW before incision, ketoprofen (non-steroidal anti-inflammatory drug) 100 mg and nefopam (non-opioid, non-steroidal, centrally acting analgesic drug) 20 mg at end of surgery. No infiltration of local anesthetic was used in the studied population cohort. Surgery was performed laparoscopically through 4 ports with a higher limit of pneumoperitoneum pressure of 12–14 cm H_2_O. After induction of anesthesia, opioids were administered according to the group in which the patient was randomized. In the SPI group, when SPI was higher than 50 for the first time for more than 3 min, and afterwards, when SPI was again higher than 50 for more than 5 min, fentanyl 1.0 mcg/kg LBW was administered, with a target SPI of 20–50. In the control arm, patients received fentanyl at the indication of the anesthesiologist, based on classical clinical signs like heart rate and blood pressure dynamics, which reflects current clinical practice.

At the end of surgery, patients were extubated following discontinuation of the volatile anesthetic and administration of neuromuscular reversal guided by TOF monitoring, to a TOF 4/4, 90%. The interval from sugammadex administration to extubation was recorded. Postoperative analgesia in the post-anesthesia care unit (PACU) consisted of tramadol administration. Rescue analgesia was administered when VAS (Visual analogue scale) score was higher than 4 points out of a maximum of 10. Assessors in the PACU were blinded regarding the allocation of patients in the control or interventional arm.

The main outcome was considered the differences between the intraoperative opioid consumption between the two groups. Secondary outcomes were the pain scores on the visual analogue scale every 15 min for the first 90 min postoperatively, time to extubation, hemodynamic events as defined below, administration of rescue analgesia.

### 2.4. Data Collection

We recorded HR, MAP, SPI at 1 min after intubation, 1 min after incision, 1 min after pneumoperitoneum, at exuflation as well as any hemodinamic events and time to extubation (defined as time from administering sugamadex to extubation), as well as total intraoperative fentanyl dose. Hemodynamic events were defined as a 20% increase from baseline of HR and MAP, as our focus was to capture nociception-related sympathetic responses, which typically manifest as tachycardia and hypertension. Decreases in HR or MAP were managed according to standard anesthetic protocols.

### 2.5. Statistical Analysis

Sample size was estimated using a preliminary study with 5 patients in each group for detecting a difference of 50 mcg of fentanyl with a significance level of 5% and statistical power of 80%. These 10 patients were included in the final analysis. We performed statistical testing using the jamovi project Version 2.6. Data are presented in Table 1, Table 2 and Table 3. Continuous variables are presented as mean ± standard deviation and categorical variables are presented as absolute numbers and percentage. Normality was assessed using the Shapiro–Wilk test. Between-group comparisons of normally distributed continuous variables were conducted with Student’s *t*-test, and non-normally distributed data with the Mann–Whitney U test. Categorical variables were compared using the χ^2^ test or Fisher’s exact test, as appropriate. A two-tailed *p* value < 0.05 was considered statistically significant.

## 3. Results

We included 49 patients in the final analysis, 24 in the SPI-guided group and 25 in the control group (Figure 1). There were no differences between the demographical data in the two groups regarding age, height, weight, BMI, gender, ASA score (Table 1). Vital signs as well as SPI recordings are detailed in Table 2. Overall, hemodynamic parameters were similar in both groups. No significant differences at any time point were registered for MAP (all *p* > 0.05) or HR, though after exsufflation there was a trend toward higher HR in the SPI group, but not significant (*p* = 0.055). A small trend toward lower SPI was registered in the SPI-guided group during incision and pneumoperitoneum (*p* = 0.066 and 0.069, respectively), despite SPI values being comparable across groups.

There were no differences regarding opioid consumption and pain scores during the first 90 min postoperatively (Table 3 and Table 4). Fentanyl dose (both absolute value and per LBW) was lower in the SPI-guided group (400 vs. 450 mcg and 7.54 ± 1.09 vs. 6.93 ± 1.52 mcg), but not statistically significant (*p* = 0.1). There was a statistically different time to wake-up between the groups of approximately 1 and 1/2 min (8.1 vs. 9.6 min, *p* < 0.001) and we also found more hemodynamic events (28% vs. 8.3%, *p* = 0.07), as well as a higher need for rescue analgesia (36% vs. 16.7%, *p* = 0.12) in the first 90 min postop in the control group, even if not statistically different. Pain scores in the first 90 min in the PACU are shown in Figure 2. In all the patients the duration of surgery was between 90 and 120 min, from induction to wake-up.

## 4. Discussion

Obesity is one of the leading causes of morbidity and mortality in the developed countries [26]. Its management consists of lifestyle changes, medical and surgical treatment [20]. The constellation of comorbidities associated with obesity raises a couple of problems for the anesthetist, such as the risk of nausea and vomiting postoperatively, the apnea–hypoventilation syndrome and susceptibility to opioid administration [27]. These risks are especially relevant when opioids are used intraoperatively, raising interest in strategies such as opioid-sparing or opioid-free anesthesia. Although there are no clear recommendations of using opioid-free anesthesia, recent evidence points to its benefits [28,29]. Our study aimed to evaluate whether using nociception monitoring during bariatric surgery reduces opioid consumption and postoperative complications related to its use.

SPI could be a more objective measure of triggering opioid administration than vital functions or clinical signs, as these can be affected by anesthetic depth, circulating volume or autonomic variability. By combining normalized heartbeat intervals and photoplethysmographic pulse wave amplitude, SPI reflects changes in sympathetic activity associated with surgical stimulation. This physiological integration provides greater specificity for nociception–antinociception balance than conventional hemodynamic monitoring alone.

Out of the 49 patients we included, 36 were female. This greater proportion of females presenting for bariatric surgery is also consistent with data from the literature [30,31] and we interpret it as them having a greater concern for both physical aspect and health.

While we found no studies in the literature focusing on nociception monitoring using SPI in obese patients, the influence of weight on SPI has been previously investigated by Chen et al., who found that SPI can accurately predict postoperative pain in patients with BMI over 25 kg/m^2^, but not in those with BMI < 25 kg/m^2^ [32].

In our cohort, vital functions (HR and MAP) were comparable at the tracked moments between the groups, with no statistically significant difference. However, there were fewer hemodynamic events, increases in HR or MAP as defined in the Materials and Methods Section, in the SPI group versus control (2/24 as opposed to 7/25). This suggests that although the overall averages of HR and MAP were similar, the frequency of acute responses to nociception were reduced when opioid administration was guided by SPI monitoring. This may be explained by opioid being administered earlier in regard to the nociceptive stimulus recognition in the SPI group, thus preventing the sympathetic surges that manifest as tachycardia or hypertension. In contrast, in the control group, opioid administration was reactive, relying on conventional hemodynamic changes, which typically lag behind the actual nociceptive stimulus. Statistical significance was not attained, possibly due to the modest sample size. Clinically, this finding is relevant: hemodynamic instability during bariatric surgery carries particular risk, as obese patients often present with cardiovascular comorbidities such as hypertension or ischemic heart disease and even transient increases in HR or MAP can increase myocardial oxygen demand and reduce hemodynamic reserve. Thus, minimizing the frequency of such events may contribute to improved perioperative stability and lower cardiovascular morbidity.

In the intervention group we targeted an SPI of 20–50 based on the results of previous studies [8,11,33]. We did not observe any significantly different SPI values between the two groups at any monitored moment, which could be attributed to the influence both laparoscopic surgery and obesity have on the autonomic nervous system physiology, and not only to the analgesia–nociception balance. SPI is an index influenced by the relationship between the sympathetic and the parasympathetic nervous system. It reflects both central (heart rate variability) and peripheral (photoplethysmographic pulse wave amplitude) sympathetic tone [19,34]. During laparoscopy, pneumoperitoneum insufflation increases sympathetic tone, thus possibly influencing SPI values by itself [34].

We found that guiding intraoperative analgesia by SPI values during bariatric surgery did not significantly reduce opioid consumption (in both absolute values of administered fentanyl and reported to the patient’s lean body weight). Data in the literature is quite heterogeneous, with some studies finding a significant reduction in opioid consumption [11,14,33,35], while in other studies doses were not reduced by guiding analgesia through SPI [34,36]. However, none of these previous studies differentiated patients according to BMI.

Despite limited impact on opioid consumption, we observed an increased speed of wake-up (9.61 min in the control group versus 8.1 min in the intervention group, *p* < 0.001). In current practice, that impact might not translate into a meaningful clinical difference. Our results are similar to other studies focusing on SPI-guided opioid administration that also found shorter times to extubation [11,14,37].

Although rescue analgesia in the first 90 min in the PACU was administered to fewer patients in the SPI group (4/24 vs. 9/25, *p* = 0.12), the difference was not statistically significant. There was also no difference in pain scores on the VAS scale in the recovery unit in the first 90 min. This could be related to the multimodal analgesia regimen administered to both groups in the perioperative period, and also to the modest sample size.

Regarding adverse effects of opioids, we did not register any episodes of apnea/desaturation postoperatively, and also no nausea/vomiting in the first 90 min postoperatively, although we did not screen for PONV after transfer from recovery room.

There is scarce data in the literature concerning nociception monitoring strictly in obese patients. Le Gall et al. [38], in their monocentric, observational, unmatched case–control study, have found that the use of ANI monitoring during bariatric surgery might reduce intraoperative consumption of opioid, although it does not appear to be accompanied by a reduction in its side effects. ANI (Analgesia nociception index) is a dimensionless score that reflects cardiac parasympathetic tone through the analysis of heart rate variability. ANI-guided analgesia resulted in a lower mean hourly consumption of sufentanil than the control group, but there was no difference between groups concerning postoperative pain scores and rescue analgesia.

### Limitations

There are several limitations regarding our research. First, the relatively modest sample size and pilot design limit the statistical power to detect small but clinically relevant differences between groups. Second, postoperative outcomes were assessed only during the first 90 min in the recovery unit. Possible differences in pain control and analgetic requirements, postoperative nausea and vomiting, or other opioid-related adverse effects beyond this time frame were not evaluated. Third, all patients received multimodal analgesia, which likely attenuated differences in opioid consumption and postoperative pain scores between groups and may have masked any measurable benefit of SPI-guided analgesia. Fourth, SPI values may be influenced by the physiological effects of pneumoperitoneum and obesity on sympathetic tone, which could limit the accuracy of nociception monitoring in this specific population. The predominance of female participants, while consistent with the demographic profile of bariatric surgery populations, also limits extrapolation of the results to males. Another limitation is that conventional fentanyl dosing at the anesthetist’s discretion may have introduced inter-provider variability, although this approach was intended to mirror real-world clinical practice rather than impose a strict dosing algorithm.

## 5. Conclusions

In our study, we found no difference in opioid requirements in patients undergoing bariatric surgery when opioids were administered guided by SPI versus by vital functions. SPI-guided analgesia might speed up emergence from general anesthesia. Nociceptive monitoring in obese patients is an area that still needs research and future studies can focus on nociception indices trends during opioid-free anesthesia.

## Figures and Tables

**Figure 1 life-15-01570-f001:**
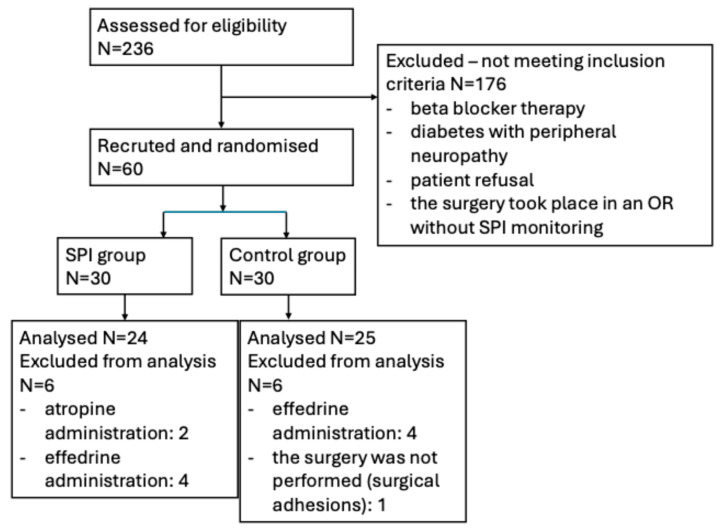
Flow diagram describing patient recruitment, randomization and withdrawal.

**Figure 2 life-15-01570-f002:**
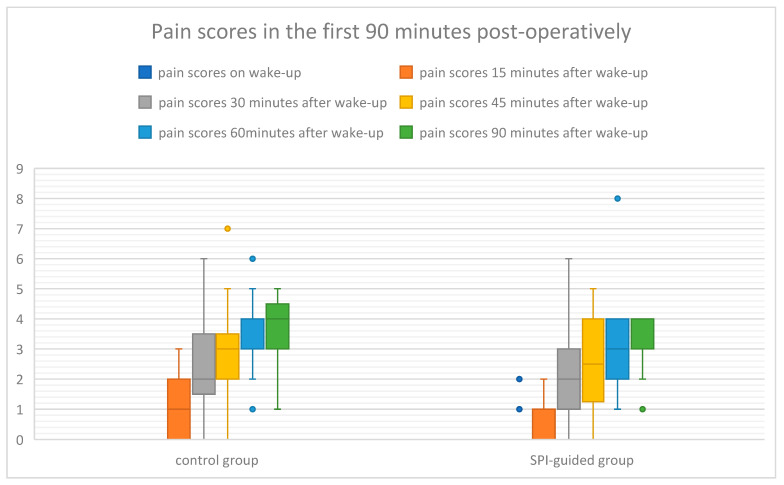
Pain scores in the first 90 min postoperatively on the visual analogue scale.

**Table 1 life-15-01570-t001:** Demographical data.

	Standard Analgesia	SPI-Guided Analgesia	*p*
Age, years (mean ± std deviation)	40 (±11.1)	38.6 (±12)	0.692
Height, cm (mean ± std deviation)	168 (±0.08)	167 (±0.08)	0.639
Weight, kg (mean ± std deviation)	122 (±21)	114 (±22.1)	0.158
Lean Body Weight, kg (mean)	58.5 (±7.66)	60.3 (±7.34)	0.802
BMI (mean ± std deviation)	43.9 (±7.07)	41.6 (±6.74)	0.410
ASA score (ASA II/ASA III/Total)	6/19/25	8/16/24	0.538
Gender (Female/Total)	20/25	16/24	0.29

**Table 2 life-15-01570-t002:** Vital functions.

		Standard Analgesia	SPI-Guided Analgesia	*p*
Mean arterial pressure (mmHg)	Preoperatively	93.60 ± 13.40	93.60 ± 13.54	0.99
	OR-admission	96.20 ± 15.37	95.60 ± 13.71	0.87
	1 min after induction	83.00 ± 9.36	88.20 ± 10.12	0.07
	1 min after incision	85.90 ± 7.72	88.40 ± 8.81	0.304
	1 min after pneumoperitoneum	86.40 ± 9.59	88.30 ± 9.85	0.481
	After exsuflation	86.30 ± 11.33	88.50 ± 10.55	0.474
Heart rate (beats/min)	Preoperatively	75 ± 10.55	79 ± 12.51	0.302
	OR-admission	78 ± 12.60	81 ± 12.38	0.596
	1 min after induction	76 ± 7.48	77.5 ± 8.41	0.489
	1 min after incision	74 ± 7.55	75.5 ± 8.83	0.077
	1 min after pneumoperitoneum	75 ± 6.76	75 ± 7.27	0.462
	After exsuflation	72 ± 8.8	77 ± 10.04	0.055
SPI	OR-admission	40.2 ± 13.49	38.5 ± 12.2	0.681
	1 min after induction	45 ± 10.74	40.8 ± 13.22	0.154
	1 min after incision	48.4 ± 7.63	43.8 ± 8.57	0.066
	1 min after pneumoperitoneum	52.7 ± 8.58	48.5 ± 6.61	0.069
	After exsuflation	51.7 ± 8.5	52 ± 5.49	0.809

**Table 3 life-15-01570-t003:** Opioid dose and time to wake-up.

	Standard Analgesia	SPI-Guided Analgesia	*p*
Fentanyl dose (mcg)	450 ± 55.9	400 ± 101.06	0.100
Fentanyl/LBW (mcg/kg)	7.54 ± 1.09	6.93 ± 1.52	0.111
Time to wake-up (min)	9.61 ± 1.34	8.1 ± 1.62	<0.001
Hemodinamic events	7/25 (28%)	2/24 (8.33%)	0.07
Rescue analgesia	9/25 (36%)	4/24 (16.66%)	0.12

**Table 4 life-15-01570-t004:** Pain scores on the visual analogue scale during the first 90 min postoperatively.

	Standard Analgesia	SPI-Guided Analgesia	*p*
Wake-up	0 ± 0	0 ± 0.448	0.153
at 15 min	1 ± 0.92	1 ± 0.72	0.889
at 30 min	2 ± 1.65	2 ± 1.5	0.682
at 45 min	3 ± 1.52	2.5 ± 1.38	0.344
at 60 min	4 ± 1.22	3 ± 1.35	0.248
at 90 min	4 ± 1.06	4 ± 0.86	0.209

## Data Availability

The raw data supporting the conclusions of this article will be made available by the authors on request.

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
