# Peer review of "Influence of Surgical Pleth Index-Guided Versus Conventional Analgesia on Opioid Consumption During Gastric Sleeve Surgery: A Pilot Study"

_life, 2025, doi:10.3390/life15101570_

Round 1
Reviewer 1 Report
Comments and Suggestions for Authors
In this paper, Leahu et al. investigate the use of SPI monitoring to guide intraoperative fentanyl dosing in obese patients undergoing bariatric laparoscopic surgery. This topic is relevant, given the challenges of opioid titration and the physiologic complexities of this patient group. The authors presented their data clearly, however I have the following concerns:
First, the control arm is problematic. “Conventional fentanyl dosing at the anesthetist’s discretion” introduces a great deal of variability, since practice style can differ widely between providers. This makes it difficult to know whether differences (or lack thereof) are due to SPI guidance or simply differences in how individual anesthetists titrated medication. The paper would be stronger if the authors either provided a standardized dosing algorithm for the control group or at least analyzed outcomes by anesthetist to account for clustering.
Secondly, surgical pain is primarily due to incisional pain, at least in the acute phase. Was local anesthetic used by the surgeon? How much and where was injected (skin vs subQ vs preperitoneal etc). It’s common practice to use local anesthetic to decrease intra and post-op pain and the authors need to comment on thatThird, did any patients need to be converted to an open procedure? If so, these patients require higher dose of narcotics for pain control.
Fourth, the manuscript does not clearly state whether an intention-to-treat analysis was performed. Were patients in the investigation arm receive fentanyl not based on SPI alone?
Fifth, were the PACU assessors blinded? If not, this can introduce a bias that needs to be reported in the limitations. What was the pain threshold to provide narcotics in the PACU and was that threshold violated at all?
Author Response
|
We thank the reviewer for summarizing our work so clearly and for recognizing the relevance of the topic. We appreciate the reviewer’s careful consideration of our study and the constructive concerns raised. Please find our detailed responses to each point below.
Comments 1: First, the control arm is problematic. “Conventional fentanyl dosing at the anesthetist’s discretion” introduces a great deal of variability, since practice style can differ widely between providers. This makes it difficult to know whether differences (or lack thereof) are due to SPI guidance or simply differences in how individual anesthetists titrated medication. The paper would be stronger if the authors either provided a standardized dosing algorithm for the control group or at least analyzed outcomes by anesthetist to account for clustering.
Response 1: We agree that “conventional fentanyl dosing at the anesthetist’s discretion” may introduce variability related to individual practice styles. Our intention was to reflect real-world clinical practice, where anesthetists often adjust dosing based on their clinical judgment rather than following a strict algorithm. In the Methods section (line 137-139), we have replaced "In the control group, fentanyl was dosed according to vital signs and the attending anesthetist" with "In the control arm, patients received fentanyl at the indication of the anesthesiologist, based on classical clinical signs like heart rate and blood pressure dynamics, which reflects current clinical practice." We acknowledge that this may limit the ability to isolate the effect of SPI guidance from inter-provider variability. To address this concern, we have added a comment in the Discussion (lines 292-295). "Another limitation is that conventional fentanyl dosing at the anesthetist’s discretion may have introduced inter-provider variability, although this approach was intended to mirror real-world clinical practice rather than impose a strict dosing algorithm."
Comments 2: Secondly, surgical pain is primarily due to incisional pain, at least in the acute phase. Was local anesthetic used by the surgeon? How much and where was injected (skin vs subQ vs preperitoneal etc). It’s common practice to use local anesthetic to decrease intra and post-op pain and the authors need to comment on that
Response 2: In order to avoid another potential source of bias, no infiltration of local anesthetic was used in the studied population cohort. We have added this information in the Methods (line 131-132) section.
"No infiltration of local anesthetic was used in the studied population cohort."
Comments 3: Third, did any patients need to be converted to an open procedure? If so, these patients require higher dose of narcotics for pain control.
Response 3: None of the patients were converted to an open procedure.
Comments 4: Fourth, the manuscript does not clearly state whether an intention-to-treat analysis was performed. Were patients in the investigation arm receive fentanyl not based on SPI alone?
Response 4: Patients in the intervention arm received fentanyl based on SPI alone.
Comments 5: Fifth, were the PACU assessors blinded? If not, this can introduce a bias that needs to be reported in the limitations. What was the pain threshold to provide narcotics in the PACU and was that threshold violated at all?
Response 5: Assesors in the PACU were blinded regarding the allocation of patients in the control or interventional arm. We have added this in the Method section (line 149-152).
"Rescue analgesia was administered when VAS (Visual analogue scale) score was higher than 4 points out of a maximum of 10. Assesors in the PACU were blinded regarding the allocation of patients in the control or interventional arm." Thank you very much for your suggestions and comentaries. |

Reviewer 2 Report
Comments and Suggestions for Authors
General comments:
- This is a very interesting manuscript for the clinical practice. Pain, although it is considered th 5th vital sign, has to be controlled in a multimodal and homogenous way. For that, the trio between obesity, pain management and opioids use is essential and can be challenging.
- This method to be applied in monitoring has to be practical and the SPI could be an important adjuvant to the monitorization of these patients, guiding intraoperative opioid administration.
- These obesity patients are a risk group due to the reduced functional residual capacity, indicating hypoventilation with more risk of apnea and respiratory arrest. This is a very interesting and new topic that should be addressed.
Specific comments:
- Introduction – Well written, structured and supported. Clear objectives.
- Material and methods:
- 1 it is not the study design;
- There should be an introductive paragraph, for example from line 93 to 93, this could be the 1stparagraph, followed by study population, and then study design, with figure 1 and explanation.
- Procedures – How was the anesthesia selection? Randomly? But how? Order of admission?
- The post operative multimodal analgesia was prescribed according to the local protocol and the attending physician. We could have a potential bias here; this should be stated as a limitation of the study;
Author Response
For research article: Influence of Surgical Pleth Index-guided versus conventional analgesia on opioid consumption during gastric sleeve surgery: a pilot study
|
Response to Reviewer 2 Comments
|
||
|
1. Summary |
|
|
|
We sincerely thank you for taking the time to review our manuscript and for your valuable feedback. Please find our detailed responses to each comment below. All corresponding revisions and corrections have been incorporated into the manuscript and are highlighted/marked in track changes in the resubmitted files.
|
||
|
2. Questions for General Evaluation |
Reviewer’s Evaluation |
Response and Revisions |
|
Does the introduction provide sufficient background and include all relevant references? |
Yes |
We have tried by the changes detailed below to improve the quality of the manuscript. |
|
Is the research design appropriate? |
Can be improved |
|
|
Are the methods adequately described? |
Can be improved |
|
|
Are the results clearly presented? |
Yes |
|
|
Are the conclusions supported by the results? |
Yes |
|
|
Are all figures and tables clear and well-presented? |
Can be improved |
|
|
3. Point-by-point response to Comments and Suggestions for Authors |
||
Comment 1: 1 it is not the study design;
Comment 2: There should be an introductive paragraph, for example from line 93 to 93, this could be the 1stparagraph, followed by study population, and then study design, with figure 1 and explanation.
Response comment 1 and 2: We have updated the Matherial and method sections according to the reviewer's suggestions. (lines 92-104)
"This randomised control trial received approval from the ethics committees of the Cluj-Napoca County Hospital (No. 57290/20.12.2022) and the University of Medicine and Pharmacy "Iuliu Hatieganu" Cluj-Napoca (No. AVZ59/12.04.2023) and was prospectively registered on clinicaltrials.gov (ID NCT05884229).
2.1. Study Population
We enrolled 49 patients, between May 2023-June 2024, undergoing laparoscopic sleeve gastrectomy in our department. All patients included in our study underwent bariatric surgery based on established clinical indications (BMI ≥ 40 kg/m2 alone or with BMI ≥ 35 kg/m2 and additional co-morbidities) and multidisciplinary evaluation at our institution. Recruitment was performed after gaining written informed consent from the patients."
2.2. Study Design
The enrolment was prospectively conducted and the flowchart is sumarised in Figure 1."
Comment 3: Procedures – How was the anesthesia selection? Randomly? But how? Order of admission?
Response comment 3: We have modified the allocation procedure in the Matherials and methos
(lines 115-118).
"Patients were randomly allocated into one of the two groups using random.org: one group with conventional opioid administration, independent of SPI, as considered by the anesthetist (the control group), and the second group with SPI-guided opioid dosing (the interventional group), as detailed further."
Comment 4: The post operative multimodal analgesia was prescribed according to the local protocol and the attending physician. We could have a potential bias here; this should be stated as a limitation of the study;
Response comment 4: We have added in the Discussion section this potential bias, as it was also suggested by one of the other reviewers. (lines 294-297)
"Another limitation is that conventional fentanyl dosing at the anesthetist’s discretion may have introduced inter-provider variability, although this approach was intended to mirror real-world clinical practice rather than impose a strict dosing algorithm."
We thank you for your kind feedback and suggestions. Please do not hesitate to contact us should there be other comments.

Reviewer 3 Report
Comments and Suggestions for Authors
Use of nociception monitoring in surgery is an area of interest but not novel. This patient group certainly offers challenges due to weight and respiratory compromise. Surgical pleth index is a method investigated in the last decade. Now there are various AI supported monitors available.
The manuscript is well written and presented.
Some comments.
Bariatric surgery in patients with mean weight of 114 - 122kg with BMI 42 appears slightly over treatment. These patients usually are advised to go for life style changes. There are 6 patients in Standard group and 8 patients in SPI group ( almost 25 -30 % of patient population) belonging to ASAII classification. This indicates that these patients had BMI under 40. Addtionaly, there were predominantly females in both groups. Although two groups do not differ(P=0.29) , predominance of female in both the group needs justification. Preliminary study for sample size needs to be mentioned in the methodology.It is apperently little late in the manuscript. Are those 5 patients included in the final sample ? This needs elaboration.
Some text needs to be modified.
Line 117 :he standard practice of noting abbreviation is to write them on first mention in a bracket after the long form and not the opposit way.
Line 119 : the word inhalatory is not a standard concept .
Line 121:IBW and PBW needs explanation.
Line 123: what is kgc? What is ketoprofen ( ketamine and propofol??) what is nefopam ??
Line 124: replace superior with higher.
Line 127-130: The reason for following questions is mentioned in discussion -why was SPI level 50 as a cut off, why for 3 -5. minutes ?what was the rational to decide 1 mcg/kg ? It needs to be expained earlier in the text to ease the logical continuety.
Line 131: what is volatile? it should be either volatile anesthetic or sevoflorane .
Line 131-133 needs to be re written.
The paragraph distribution in the lines from 131-139 should be reconsidered.
Line 134: The protocol for postoperarive multimodal analgesia needs to mentioned.
Line 136: " intraoperative "is missing , I guess.
Line 138: why only up to 90 minutes? 15 minutes is too frequent. It takes almost 10-15 minutes for any analgesia to show effect after admnistration. The administration it self takes 5 minutes .
Line 141 : needs rewriting. The present text means total fentanyl dose was recorded at the mentioned timepoints.
Line 144: Why only increase ?
In statistics, the weight is described in mean and SD. was the data in both the group normally distributed ?
Table 2. The tables need to be re arranged.
The mean MAP appear quite high . Patients were young (mean age 38-40)and not hypertensive.
Table 3. Sugamedex dose is not shown.
Table4. VAS at 90 minute ??? why is it not dísplayed?
Line 200: typo. vital functions.
In Discussion, the advantage of SPI over standard hemodynamic parameters needs to be elaborated with little physiological explanation.
Author Response
|
We thank the reviewer for this thoughtful comment and for acknowledging the clarity of our manuscript.
Comments 1: Bariatric surgery in patients with mean weight of 114 - 122kg with BMI 42 appears slightly over treatment. These patients usually are advised to go for life style changes. There are 6 patients in Standard group and 8 patients in SPI group ( almost 25 -30 % of patient population) belonging to ASAII classification. This indicates that these patients had BMI under 40. Addtionaly, there were predominantly females in both groups. Although two groups do not differ(P=0.29) , predominance of female in both the group needs justification. Preliminary study for sample size needs to be mentioned in the methodology.It is apperently little late in the manuscript. Are those 5 patients included in the final sample ? This needs elaboration. Response 1: We thank the reviewer for these detailed and thoughtful observations. Please find our responses below: · Patient weight/BMI and indication for bariatric surgery. According to the 2022 American Society of Metabolic and Bariatric Surgery (ASMBS) and International Federation for the Surgery of Obesity and Metabolic Disorders (IFSO) Indications for Metabolic and Bariatric Surgery, indications for bariatric surgerry extend to BMI ≥ 35 kg/m2 and an obesity-related comorbidity that could be expected to improve/resolve by surgery-induced weight loss maintenance, BMI ≥ 40 kg/m2 alone or type-2 diabetes mellitus (T2D) in patients with class-I obesity and inadequately controlled hyperglycaemia despite optimal medical treatment.We agree that lifestyle changes are generally the first-line recommendation for patients with obesity. However, all patients included in our study underwent bariatric surgery based on established clinical indications and multidisciplinary evaluation at our institution. · ASA II classification and BMI under 40: We acknowledge the reviewer’s observation that a subset of patients fell into ASA II classification and had BMI values below 40. These patients were nonetheless scheduled for bariatric surgery due to additional comorbidities and clinical judgment, consistent with accepted practice in our center. We have clarified this point in the Methods section (lines 96-99). "All patients included in our study underwent bariatric surgery based on established clinical indications (BMI ≥ 40 kg/m2 alone or with BMI ≥ 35 kg/m2 and additional co-morbidities) and multidisciplinary evaluation at our institution." · Predominance of females in both groups: The predominance of female patients reflects the demographic profile of individuals seeking bariatric surgery, consistent with broader epidemiological data reporting a higher proportion of females undergoing such procedures, as we have explained in the Discussion. · Sample size calculation: We have moved the information regarding the preliminary sample size description to the Methods section (lines 107-110) to ensure clarity. "For sample size estimation we preliminarily enrolled 5 patients in each group. We used the data for calculating the sample size for detecting a difference of 50 mcg of fentanyl with a significance level of 5% and statistical power of 80%." · Inclusion of the 5 patients: We confirm that the 5 patients referenced were included in the final analysis. We have clarified this explicitly in the Methods (line 157) to avoid any ambiguity. "These 10 patients were included in the final analysis."
Some text needs to be modified. Response line 117: We have changed the abbreviations between the brackets. " train-of-four (TOF), bispectral index (BIS) "
Line 119 : the word inhalatory is not a standard concept . Response line 119: We dropped the word 'inhalatory' Line 121:IBW and PBW needs explanation. Response line 121: We explained between paranthesis the 2 abbreviations. Line 123: what is kgc? What is ketoprofen ( ketamine and propofol??) what is nefopam ?? Response Line 123: We have corrected kgc to kg on line 123. Ketoprofen is a non-steroidal inflamatory drug and nefopam is is a non-opioid, non-steroidal, centrally acting analgesic drug. We have modified the the text to make this more clear. "ketoprofen (non-steroidal anti-inflamatory drug) 100 mg and nefopam (non-opioid, non-steroidal, centrally acting analgesic drug)" Line 124: replace superior with higher. Response line 124: We replaced 'superior' with 'higher'.
Response line 127-130: We have rephrased in the Introduction lines 59-60 ("The cut-off of 50 for a period more than 3-5 minutes has been previously used as threshold to administer further doses of opioid during general anesthesia") to enhance the logical continuity with "A threshold value of 50 sustained for more than 3–5 minutes has previously been used as an indication to administer additional opioid doses during general anesthesia". We selected an intraoperative bolus dose of 1 µg/kg fentanyl because this dose lies within the range commonly used in anesthetic practice for titrating analgesic effect while mitigating the risks of oversedation or respiratory depression.
Response line 131: We have replaced 'volatile' with 'volatile anesthetic'
Line 134: The protocol for postoperative multimodal analgesia needs to mentioned. Response line 131-139: We have re-written and re-organised the lines 131-139 as follows. We have also added information regarding postoperative analgesia plan, as suggested by the reviewer. "At the end of surgery, patients were extubated following discontinuation of the volatile anesthetic and administration of neuromuscular reversal guided by TOF monitoring, to a TOF 4/4, 90%. The interval from sugammadex administration to extubation was recorded. Postoperative analgesia in the PACU consisted of tramadol administration. Rescue analgesia was administered when VAS (Visual analogue scale) score was higher than 4 points out of a maximum of 10. Assesors in the PACU were blinded re-garding the allocation of patients in the control or interventional arm. " (lines 141-147)
Line 136: " intraoperative "is missing , I guess. Response line 136: We have added "intraoperative opioid consumption". Line 138: why only up to 90 minutes? 15 minutes is too frequent. It takes almost 10-15 minutes for any analgesia to show effect after admnistration. The administration itself takes 5 minutes .
Line 141: needs rewriting. The present text means total fentanyl dose was recorded at the mentioned timepoints. Response line 141: We have rewritten the phrase to make it more clear. We recorded HR, MAP, SPI at 1 minute after intubation, 1 minute after incision, 1 minute after pneumoperitoneum, at exuflation as well as any hemodinamic events and time to extubation (defined as time from administering sugamadex to extubation), as well as total intraoperative fentanyl dose.(153-156)
Response line 144: We considered only increases in HR and MAP as hemodynamic events only for the purpose of the study because our focus was to capture nociception-related sympathetic responses, which typically manifest as tachycardia and hypertension. A decrease in HR or MAP is more often attributable to anesthetic depth, fluid status, or other perioperative factors, and thus was not included in our predefined nociception-related outcomes. We agree that hypotension and bradycardia are clinically relevant events and they were managed according to standard anesthetic protocols. We have added this clarification to the Methods (line 156-159). "Hemodynamic events were defined as a 20% increase from baseline of HR and MAP, as our focus was to capture nociception-related sympathetic responses, which typically manifest as tachycardia and hypertension. Decreases in HR or MAP were managed according to standard anesthetic protocols."
Response: Could you please explain how to re-arrange the tables?
Table 3. Sugamadex dose is not shown. Response: Sugamadex dose was adjusted for a target of TOF (4/4, >90%). We did not report it, as it does not have any influence on the study's outcomes.
Response Table 4: We have added the values of pain scores at 90 minutes to the table. Table 4. Pain scores on the visual analogue scale during the first 90 minutes post-operatively
Line 200: typo. vital functions. Response line 200: We have dropped the word "values".
In Discussion, the advantage of SPI over standard hemodynamic parameters needs to be elaborated with little physiological explanation. Response: We have elaborated on the advantages of SPI over standard hemodynamic parameters in the Discussions. (line 208-214) " SPI could be a more objective measure of triggering opioid administration than vital functions or clinical signs, as these can be affected by anesthetic depth, circulating volume or autonomic variability. By combining normalized heart beat intervals and photople-thysmographic pulse wave amplitude, SPI reflects changes in sympathetic activity as-sociated with surgical stimulation. This physiological integration provides greater spec-ificity for nociception–antinociception balance than conventional hemodynamic moni-toring alone." Thank you very much for your suggestions. Should there be other comments please do not hesitate to contact us further. |

Round 2
Reviewer 1 Report
Comments and Suggestions for Authors
I’d like to thank the authors for their response. No further concerns.